# Phenothiazines Rapidly Induce Laccase Expression and Lignin-Degrading Properties in the White-Rot Fungus *Phlebia radiata*

**DOI:** 10.3390/jof9030371

**Published:** 2023-03-18

**Authors:** Matthew P. Hirakawa, Alberto Rodriguez, Mary B. Tran-Gyamfi, Yooli K. Light, Salvador Martinez, Henry Diamond-Pott, Blake A. Simmons, Kenneth L. Sale

**Affiliations:** 1Systems Biology Department, Sandia National Laboratories, Livermore, CA 94550, USA; 2Biomaterials and Biomanufacturing Department, Sandia National Laboratories, Livermore, CA 94550, USA; 3Bioresource and Environmental Security Department, Sandia National Laboratories, Livermore, CA 94550, USA; 4Biological Systems & Engineering Division, Lawrence Berkeley National Laboratory, Berkeley, CA 94720, USA; 5Deconstruction Division, Joint BioEnergy Institute, Emeryville, CA 94608, USA; 6Computational Biology and Biophysics Department, Sandia National Laboratories, Livermore, CA 94550, USA

**Keywords:** white-rot fungi, laccase, lignin, phenothiazine

## Abstract

*Phlebia radiata* is a widespread white-rot basidiomycete fungus with significance in diverse biotechnological applications due to its ability to degrade aromatic compounds, xenobiotics, and lignin using an assortment of oxidative enzymes including laccase. In this work, a chemical screen with 480 conditions was conducted to identify chemical inducers of laccase expression in *P. radiata*. Among the chemicals tested, phenothiazines were observed to induce laccase activity in *P. radiata,* with promethazine being the strongest laccase inducer of the phenothiazine-derived compounds examined. Secretomes produced by promethazine-treated *P. radiata* exhibited increased laccase protein abundance, increased enzymatic activity, and an enhanced ability to degrade phenolic model lignin compounds. Transcriptomics analyses revealed that promethazine rapidly induced the expression of genes encoding lignin-degrading enzymes, including laccase and various oxidoreductases, showing that the increased laccase activity was due to increased laccase gene expression. Finally, the generality of promethazine as an inducer of laccases in fungi was demonstrated by showing that promethazine treatment also increased laccase activity in other relevant fungal species with known lignin conversion capabilities including *Trametes versicolor* and *Pleurotus ostreatus.*

## 1. Introduction

White-rot fungi are a diverse and widespread collection of basidiomycete species that can degrade all structural components of the plant cell wall, including lignin [1]. In nature, white-rot fungi are the most efficient lignin-degrading organisms on the planet and play key roles in the global carbon cycle [2,3]. To degrade lignin, white-rot fungi produce a battery of non-specific extracellular oxidative enzymes that primarily belong to four classes: lignin peroxidases, manganese peroxidases, versatile peroxidases, and laccases [4,5,6]. Together, these ligninolytic enzymes are capable of depolymerizing diverse high-molecular-weight recalcitrant lignin heteropolymers into smaller aromatic products that are funneled into central carbon metabolism of white-rot species [7]. Lignin-degrading enzymes are also capable of degrading related aromatic compounds found in polycyclic aromatic hydrocarbons, pesticides, synthetic dyes, explosives, and synthetic polymers, making white-rot fungi promising candidates for bioremediation applications [8,9,10].

Lignin degrading enzymes in fungi are encoded by numerous gene families that each contain multiple distinct isoenzymes [5,11,12]. The regulation of ligninolytic enzyme expression has mostly been studied in the model white-rot species *Phanerochaete chrysosporium*, and early studies demonstrated that nutrient limitations induce the expression of ligninolytic and carbohydrate-active enzymes through secondary metabolic processes [13,14]. In addition to nutrient limitation, an assortment of chemical inducers, carbon sources, and environmental stressors can also increase the expression of ligninolytic enzymes in white-rot fungi [11,15,16,17]. However, the regulation of ligninolytic enzyme expression varies significantly across species, gene families and enzyme isoforms, making it difficult to generalize the conditions that induce lignin-degrading properties across white-rot fungi [11]. Together, the current data suggest that the expression of ligninolytic enzymes in white-rot fungi is controlled by multiple signaling and regulatory programs in response to diverse environmental stimuli.

Laccases are a family of multi-copper containing oxidoreductases that catalyze the oxidation of diverse phenolic substrates [4,18,19]. Because of their broad substrate range, laccases are particularly versatile enzymes for use in diverse biotechnological industries and applications including bioremediation, biofuels, textiles, food additives, biosensors, and pulp and paper production [19,20,21]. In nature, laccases are found across the tree of life in species of insects, plants, bacteria, and fungi, where they serve many purposes including stress defense, morphogenesis, host–pathogen interactions, and lignin degradation [22]. Interestingly, although laccases are widely distributed among fungi, these enzymes are absent in the model white-rot species *P. chrysosporium* [3,12]. Instead, most research regarding laccase in white-rot fungi has been conducted using *Cerrena unicolor*, *Dichomitus squalens*, *Pleurotus spp*., *Phlebia spp.*, and *Trametes versicolor* [23,24,25,26,27,28]. Laccase expression in basidiomycete fungi is usually low in replete media, but altering growth conditions can increase the expression of these extracellular enzymes [29]. The transcriptional regulation and expression of laccase genes in fungi is controlled by specific environmental conditions and nutrients including metal ions, nitrogen and carbon sources, and various aromatic compounds [24,30,31,32]. 

Among the most potent laccase producing fungi are species belonging to the *Phlebia* genus [26]. *Phlebia radiata* is a particularly intriguing species with a high potential for biotechnological use due to its ability to produce ethanol from lignocellulosic biomass and its ability to degrade lignin, lignin-like aromatic compounds, and xenobiotics [33,34]. Additionally, the *P. radiata* genome encodes an extensive repertoire of carbohydrate-active enzymes (CAZymes) and lignin modifying enzymes [35]. The expression of lignin-degrading enzymes in *P. radiata* can be enhanced through specific culture conditions that include growth on wood, copper, nitrogen supplementation, and exposure to aromatic compounds [36,37,38,39,40]. Although several conditions have been identified that increase lignin-degrading enzyme activity in *P. radiata*, the molecular mechanisms regulating the expression of these enzymes in *P. radiata* remains relatively understudied. 

In this study, a chemical screen was conducted to identify compounds that induce laccase expression in *Phlebia radiata.* Through this work, several phenothiazine compounds were identified that rapidly and potently increased laccase expression in *P. radiata*. In particular, promethazine, a compound typically used as an anti-psychotic drug in humans, altered transcriptomes of *P. radiata* by increasing the expression of genes associated with lignin degradation and xenobiotic metabolism. Additionally, promethazine enhanced the ability of *P. radiata* to degrade phenolic model lignin compounds. Together, this study identified phenothiazines as chemical inducers of lignin-degrading properties in white-rot fungi.

## 2. Materials and Methods

### 2.1. Strains and Culture Conditions

The fungal species used in this study were *Phlebia radiata*, *Bjerkandera adusta*, *Dichomitus squalens*, *Phanerochaete chrysosporium*, and *Pleurotus ostreatus*, and were acquired from ATCC (ATCC, Manassus, VA, USA). Natural isolates of *Trametes versicolor* used in this study were purified from fruiting bodies collected in Pleasanton, CA, USA and Tracy, CA, USA, and a natural isolate of *Laetiporus gilbertsonii* was purified from a fruiting body collected in Pleasanton, CA. Species-level identification of natural fungal isolates was conducted by PCR amplification of the ITS region from genomic DNA using primers NSI1 (GATTGAATGGCTTAGTGAGG) and NLB4 (GGATTCTCACCCTCTATGAC) [41]. PCR amplicons were subcloned into TOPO-TA vectors (Invitrogen, Waltham, MA, USA), sequenced by Sanger sequencing (Azenta Life Sciences, Chelmsford, MA, USA) and aligned to sequence databases using nucleotide BLAST (NIH NCBI, Bethesda, MD, USA). Fungal mycelia used in this study were routinely cultured on Potato Dextrose Agar (PDA) at 25 °C (BD Difco, Franklin Lakes, NJ, USA). 

### 2.2. Chemical Screen Using Phenotype Microarrays

To conduct phenotype microarray (PM) experiments with *P. radiata*, methods were adapted from protocols provided by Biolog Inc. and a previous study that utilized PM plates for basidiomycete fungi in the order Polyporales [42]. Protocols were optimized here to ensure that equal inoculums were distributed to each well of the PM plates. *P. radiata* mycelia were removed from the surface of PDA plates using a sterile cotton swab and resuspended in liquid Yeast Nitrogen Base (YNB) + 2% D-glucose liquid media (BD Difco). To prepare a homogenous inoculum, mycelia suspensions were vigorously vortexed and subsequently passed through a 40 µm nylon cell strainer to remove large mycelial aggregates and enrich for small mycelial fragments. The densities of filtered cell suspensions were quantified using OD_600_ measurements, and suspensions were vortexed and diluted to a final concentration of OD_600_ = 0.05 in YNB + 2% D-glucose media. Phenotype microarray plates designed for chemical sensitivity tests for fungi (PM21–PM25) were inoculated with 100 µL of diluted cell suspensions per well (Biolog Inc., Hayward, CA, USA). PM plates were incubated at 25 °C in the dark without agitation for 7 days. To quantify cell growth in the presence of different chemical stressors, PM plates were analyzed using a Tecan Spark plate reader to collect OD_600_ values (Tecan Group Ltd., Männedorf, Switzerland). To measure laccase activity in phenotype microarray experiments, ABTS oxidation assays were utilized as described in the section below. 

### 2.3. Promethazine Induction Experiments

For bench-scale promethazine induction experiments, equal-sized fungal mycelia plugs were excised from PDA plates using 8.0 mm biopsy punches (Ted Pella Inc., Redding, CA, USA) and inoculated into 20 mL liquid YNB + 2% glucose media. Liquid cultures were incubated on a shaking platform for 5 days at 30 °C with agitation set to 80 RPM to allow growth. Following the 5-day outgrowth, liquid cultures were treated with promethazine hydrochloride (Thermo Scientific, Waltham, MA, USA) to a final concentration of 10 µg/mL (accounting for 1/1000 of the total culture volume) and returned to the 30 °C shaking incubator for the duration of each experiment. Media supernatants or fungal mycelia were collected at multiple time points defined throughout the manuscript and were analyzed using approaches described in sections below. The same protocol was used to treat fungal mycelia with chlorpromazine, phenothiazine, veratryl alcohol, copper sulfate, and lignin. 

### 2.4. Laccase Activity Assays

Quantitative laccase activity assays were adapted from previously established methods using ABTS oxidation measurements [43,44]. Laccase activity assays were performed in 200 µL reaction volumes containing 0.5 mM ABTS, 20 µL fungal culture supernatant, and 100 mM potassium phosphate buffer with pH 5. For ABTS assays performed using fungal secretomes collected from phenotype microarray plates, reaction mixtures were incubated at room temperature for 20 min. For all other ABTS assays in this study, reaction mixtures were incubated at room temperature for 5 min. Immediately following incubation, absorbance was measured at 420 nm (*ε*_420_ = 36,000 M^−1^ cm^−1^; *d* = 1 cm) using a Tecan Spark plate reader. One unit of enzyme activity was defined as the amount of enzyme oxidizing 1 µmol of substrate min^−1^. Numerous formulas have been reported in the literature to calculate laccase enzymatic activity; here, we utilized the formula below because it was recently reported to be the most appropriate [44].
(1)$${UL}^{-1}=(∆A\times Vt\times Df\times 10^{6})/(t\times \varepsilon \times d\times Vs)$$

*ΔA* = final absorbance − initial absorbance; *Vt* = total reaction volume (mL), *Df =* dilution factor, *t =* time (min), *ε* = molar extinction coefficient (M^−1^ cm^−1^), *d* = optical path (cm), vs. = sample volume (mL). Data for laccase activity assays were analyzed and plotted using GraphPad Prism 9 software.

### 2.5. Secretome Analysis Using Label-Free Quantitative Proteomics

*P. radiata* mycelia were grown as described above for promethazine induction experiments, and media supernatants were collected in biological triplicates 2 days after treatment. *P. radiata*-conditioned media supernatants (~20 mL each sample) were flash frozen using liquid nitrogen and sent for sample preparation and secretome analysis using label-free quantitative proteomics (Creative Proteomics, Shirley, NY, USA). The total protein of each sample was precipitated using methanol and chloroform and the protein pellet was dissolved in a 2 M urea aqueous solution. Samples were then denatured with 10 mM DL-dithiothreitol at 56 °C for 1 h followed by alkylation at room temperature for 1 h in the dark using 50 mM iodoacetamide. Next, ammonium bicarbonate was added to the solution to a final concentration of 50 mM at pH 7.8, and samples were then digested with trypsin (Promega, Madison, WI, USA) at 37 °C for 15 h. Peptides were further purified with C18 SPE column (Thermo Scientific, Waltham, MA, USA) to remove salt, lyophilized to near dryness, and then resuspended in 20 μL of 0.1% formic acid. 

LC-MS/MS was performed by loading 1 μg sample on the Ultimate 3000 nano UHPLC system (Thermo Fisher Scientific, Waltham, MA, USA) with trapping column (PepMap C18, 100Å, 100 μm × 2 cm, 5 μm) and an analytical column (PepMap C18, 100Å, 75 μm × 50 cm, 2 μm). Mobile phase consisted of buffer A (0.1% formic acid in water) and buffer B (0.1% formic acid in 80% acetonitrile), using a total flow rate of 250 nL/min and the following LC linear gradient: from 2% to 8% buffer B in 5 min, from 8% to 20% buffer B in 60 min, from 20% to 40% buffer B in 32 min, and then from 40% to 90% buffer B in 4 min. Mass spectrometry full scan was performed between 300–1650 m/z at the resolution 60,000 at 200 m/z, the automatic gain control target for the full scan was set to 3 × 10^6^. The MS/MS scan was operated in top-20 mode using the following settings: resolution 15,000 at 200 m/z; automatic gain control target 1 × 10^5^; maximum injection time 19ms; normalized collision energy at 28%; isolation window of 1.4 Th; charge sate exclusion: unassigned, 1, >6; and dynamic exclusion 30 s. 

Raw MS files were analyzed and searched against the Uniprot *Phlebia radiata* protein database using Maxquant (1.6.2.14). The parameters were set as follows: the protein modifications were carbamidomethylation (C) (fixed) and oxidation (M) (variable), the enzyme specificity was set to trypsin, the maximum missed cleavages were set to 2, the precursor ion mass tolerance was set to 10 ppm, and the MS/MS tolerance was 0.5 Da. 

### 2.6. Degradation Analysis of Lignin Model Compounds

*P. radiata*-conditioned media supernatants were collected at four time points after promethazine treatment (2 h, 1 day, 2 days, and 7 days) and filtered through 40 µm cell strainers to remove mycelia. Degradation reactions were performed in 200 µL volumes containing 20 µL filtered fungal culture supernatant and 180 µL of a lignin model compound diluted to 200 µg/mL in 100 mM sodium acetate buffer with pH 5. The dimeric lignin model compounds used in this study were 5,5′-dehydrodivanillate (DDVA), guaiacylglycerol-β-guaiacyl ether (GGE), syringylglycerol-β-guaiacyl ether (SGE), and veratrylglycerol-β-guaiacyl ether (VGE) (GreenLignol, LLC., San Francisco, CA, USA). Degradation reactions were incubated at 25 °C for 3 days without agitation and then filtered through 0.45 µm filters prior to HPLC analysis. 

To quantify the abundance of lignin model compounds, filtered samples were analyzed with an Agilent 1290 Infinity II HPLC system equipped with a UV detector and an Eclipse Plus Phenyl-Hexyl column maintained at 50 °C (Agilent Technologies, Santa Clara, CA, USA) using a previously reported method [45]. The mobile phase utilized a gradient profile that included 10 mM ammonium acetate and 0.07% formic acid in water (solvent A) and 10 mM ammonium acetate and 0.07% formic acid in 90% acetonitrile (solvent B) in varying ratios. The gradient profile used included the following solvent A/B% ratios is as follows: 70/30 (0 min, 0.5 mL/min); 20/80 (12 min, 0.5 mL/min); 0/100 (12.1 min, 0.5 mL/min); 0/100 (12.6 min, 1.0 mL/min); 70/30 (12.8 min, 1.0 mL/min); and 70/30 (15.6 min, 1.0 mL/min). The percentage of compound degradation was determined by quantification of the reduction in peak height detected at 280 nm of lignin model compounds incubated with *P. radiata*-conditioned media compared to their respective pure standards. 

### 2.7. RNA Purification, RNA-Sequencing and Transcriptomic Analyses

*P. radiata* mycelia were scraped off PDA plates using a sterile swab and resuspended in liquid YNB + 2% glucose media. Mycelia were filtered through a 40 µm cell strainer to remove large cell clumps, cell density measured at OD_600_, and subsequently diluted into 20 mL fresh YNB + 2% glucose to a final concentration of OD_600_ = 0.001. Fungal cell cultures were then grown in flasks on a shaking platform incubator at 30 °C with agitation at 100 RPM for 7 days. After 7 days, cells were treated with promethazine at a final concentration of 10 µg/mL, or with an equivalent volume of water for control samples (volumes of treatments represented 1/1000 of the total culture volume). Samples were harvested at 2 h, 24 h, and 48 h post treatment by centrifuging the contents of the flasks at 5000 RCF for 5 min at 4 °C in 50 mL conical tubes, removing the media supernatant, and flash freezing cell pellets in liquid nitrogen. Each sample was grown and harvested in biological triplicate. Frozen cell pellets were stored at −80 °C until further processing. 

To isolate total RNA from frozen fungal pellets, ~50 mg of frozen mycelia was first resuspended in 750 µL Qiagen RLT Buffer + 1% β-mercaptoethanol in 2 mL tubes containing Lysing Matrix Y zirconium oxide beads (MP Biomedicals, Irvine, CA, USA). Mycelia were vortexed at maximum speed for 10 s, flash frozen using liquid nitrogen, and then stored at −80 °C. Next, samples were thawed on ice and lysed using a FastPrep-24 bead beater for 25 s at 40 Hz (MP Biomedicals). Samples were then centrifuged at 9000 RCF for 10 min at 4 °C to pellet zirconium beads and cell debris, and RNA was purified from supernatants using the Qiagen RNeasy Plant Mini Kit as per the manufacturer’s instructions (Qiagen Sciences, Germantown, MD, USA). Purified RNA was quantified using a NanoDrop 2000 (Thermo Fisher Scientific, Waltham, MA, USA), and RNA integrity was measured using a 4200 TapeStation System with RNA ScreenTape analysis (Agilent Technologies). 

RNA-seq libraries were generated using the KAPA mRNA HyperPrep Kit with KAPA Unique-Dual Indexed adapters (Roche Sequencing and Life Sciences, Wilmington, MA, USA). RNA sequencing was performed using the Illumina NextSeq 500/550 platform with the High Output v2 kit (150 cycles) on paired-end mode (Illumina Inc., San Diego, CA, USA). BCL files were converted to FASTQ and demultiplexed using the bcl2fastq conversion software (Illumina, Inc.). Quality filtering and adaptor trimming were performed using fastp with following parameters: qualified_quality_phred 25, cut_window_size 5–3, and cut_mean_quality 25 [46]. Transcript abundance was quantified from sequencing reads using the Kallisto pseudoaligner with a *P. radiata* transcriptome index built from transcript annotation files acquired from the JGI MycoCosm website [37,47,48]. Transcript IDs and gene annotations were assigned to gene models using the *P. radiata* genome database on MycoCosm [37,48]. Principal component analysis and differential expression analysis of RNA-seq data were performed using Sleuth [49]. GO term analysis was performed using Blast2GO [50]. Results were analyzed and plotted using RStudio and GraphPad Prism software.

## 3. Results

### 3.1. Chemical Screen to Identify Compounds That Induce Laccase Expression in P. radiata

The expression of lignin-degrading enzymes in fungi is controlled by diverse environmental cues [11]. Laccase expression in fungi is typically low, but growth in specific environmental conditions can increase the expression of these enzymes [29,30]. Here, the aim was to identify novel chemical inducers of laccase expression in *P. radiata* using a high-throughput chemical screen coupled with laccase activity assays and transcriptomics analysis. Using Biolog Phenotype Microarray (PM) plates designed for fungal sensitivity assays (Appendix A), the ability of *P. radiata* to grow and produce laccase was measured across 480 conditions tested in parallel. *P. radiata* sensitivity to chemical stressors was first analyzed by quantifying growth of fungi across PM plate conditions after 7 days growth at 25 °C using OD_600_ measurements (Figure 1A). *P. radiata* was most sensitive to compound 48/80, trifluoperazine, 3-amino-1,2,4-triazole, polymyxin B, azaserine, paromomycin, thialysine, cisplatin, hydroxyurea, amitriptyline, alexidine, 5-fluorodeoxyuridine, and 5-fluorouracil and was not capable of growth when exposed to these chemicals under any concentration tested. 

To identify chemicals that increased laccase activity in *P. radiata*-conditioned media, filtered media supernatants were collected from PM plates after 7 days growth at 25 °C and were individually tested for their ability to oxidize 2,2′-azino-bis(3-ethylbenzothiazoline-6-sulfonic acid) (ABTS) (Figure 1A,B). The chemicals that induced the highest levels of secreted laccase activity by *P. radiata* included berberine chloride, promethazine, zaragozic acid A, chlorpromazine, and sodium salicylate. To test whether the laccase activity observed was a function of total biomass, laccase activity was plotted against cell density and analyzed for correlation. Laccase activity and cell growth were not correlated (*R*^2^ = 0.11), indicating that increased levels of laccase production are not resulting directly from an increase in biomass. The chemical structures of the top-five laccase-inducing compounds indicated that all were aromatic compounds, which is consistent with previous studies investigating lignin-degrading properties in *P. radiata* (Figure 1C) [39,40]. Among the top 20 strongest laccase-inducing compounds, four belonged to a class of phenothiazines that are used for anti-psychotic therapies in humans (promethazine, chlorpromazine, thioridazine, and trifluoperazine). To our knowledge, phenothiazines have not been previously identified as inducers of lignin-degrading activity in fungi, and therefore, the response of *P. radiata* to these compounds was further investigated in this study.

### 3.2. Promethazine as a Rapid and Potent Inducer of Laccase Activity in P. radiata

Biological processes in fungal systems are often inconsistent across culture conditions, especially during scale-up experiments. To examine whether promethazine (the strongest phenothiazine derived laccase inducer in the chemical screen) exposure induced laccase expression in bench-scale cultures, laccase activity was measured in culture supernatants when fungal culture volumes were scaled-up 200× from the Biolog PM plate chemical screen (100 µL to 20 mL culture volumes). To identify the optimal laccase-inducing concentrations of promethazine, laccase activity was measured after growing *P. radiata* in YNB + 2% glucose media with promethazine concentrations ranging from 0 ug/mL to 1000 ug/mL. The optimal laccase-inducing concentration for *P. radiata* was observed at 10 µg/mL, and this concentration was used for all subsequent experiments in this study (Appendix A). When *P. radiata* was exposed to promethazine, a 21× increase in laccase activity over the untreated controls was observed (*p* = 6.2 × 10^−9^, *t*-test) (Figure 2A). Chlorpromazine (the second-strongest phenothiazine derivative laccase inducer observed) also increased laccase activity by 3× when compared to untreated controls in scale-up experiments (*p* = 0.0095, *t*-test). 

Because multiple phenothiazine derivatives contained in the chemical screen were observed to increase the extracellular laccase activity of *P. radiata*, the ability of the phenothiazine backbone itself to induce laccase activity in this species was also examined (Appendix A). Similar to the promethazine treatment, when *P. radiata* was treated with phenothiazine, extracellular laccase activity in the media increased by 25× in comparison to the untreated controls (*p* = 7.5 × 10^−5^, *t*-test). Together, these results demonstrate that treating *P. radiata* with phenothiazine compounds increased extracellular laccase activity in bench-scale culture systems. 

To investigate the temporal dynamics of extracellular laccase activity in response to promethazine, laccase activity when *P. radiata* was exposed to promethazine in synthetic and rich media over time was examined (Figure 2B,C). In these experiments, media samples were collected at different time points over the course of 14 days and analyzed for laccase activity. In both media types, an increase in laccase activity was observed starting at 1 day post promethazine treatment and peaking ~2–4 days post treatment (note: enzyme activity values for PMZ-treated samples in YNB cultures on days 2–4 may be underrepresented due to the saturation of ABTS substrate). Interestingly, in both media types, laccase activity declined at 7 days post treatment. This decline in laccase activity could be delayed with additional promethazine treatments over time, suggesting that promethazine may be degraded during the experiment (Appendix A). When *P. radiata* was grown in synthetic media, laccase activity rebounded at the 10- and 14-day time points. Alternatively, when *P. radiata* was cultured in rich media, laccase activity continued to decline until the end of the experiment. In both media types, untreated control *P. radiata* cultures were observed to gradually increase laccase activity over time, with the highest levels of laccase activity occurring on the final 14-day time point. Together, these results demonstrate that promethazine functions as a potent and rapid, but transient, inducer of laccase activity in *P. radiata*. 

Promethazine induction was next compared to several other previously described laccase-inducing compounds including copper sulfate, veratryl alcohol, and lignin derived from corn stover (Appendix A) [36,38,39]. In these experiments, *P. radiata* mycelia were grown in 20 mL YNB + 2% glucose and treated with either CuSO_4_ (150 μM), veratryl alcohol (2 mM), an autoclaved solution of lignin derived from corn stover (50 μg/mL), promethazine (10 μg/mL), or given no treatment as a control. Conditioned media supernatants from *P. radiata* were analyzed for laccase activity at several time points over the course of 1 week. At 2 days post treatment, *P. radiata* cultured with a lignin solution derived from corn stover induced a slight ~1.5× increase in laccase activity compared to the control (*p* = 0.001, *t*-test). When *P. radiata* was treated with veratryl alcohol, a ~7× increase in laccase activity compared to controls (*p* = 0.007, *t*-test) was observed. Unlike promethazine treatment, laccase activity did not decline and continued to increase over the course of the week when *P. radiata* was treated with veratryl alcohol. No change in laccase activity was observed when *P. radiata* was treated with CuSO_4_; however, previous studies used higher concentrations of this compound and found that laccase induction occurs at later time points than those tested here [36]. Overall, of the conditions tested, promethazine induced the most rapid and highest levels of laccase activity.

Although ABTS oxidation assays are commonly used to measure laccase activity, ABTS is not a selective substrate for laccase and it is possible that other secreted enzymes may be involved with its oxidation. Therefore, to examine how promethazine treatment alters the secretome of *P. radiata*, label-free quantitative proteomics analysis was performed on *P. radiata* media supernatants 2 days after treatment with promethazine. Using this approach, the only differentially expressed protein identified was laccase (*lac1*), and the abundance of this protein increased by 14.5× in response to promethazine (*p* = 1 × 10^−4^) (Figure 2D). Other proteins identified in the media were two lignin peroxidases (*lgp3* and *lip4*), two manganese peroxidases (*mnp2* and *mnp3*), a translation elongation factor EF1-alpha (*tef1*), and an uncharacterized protein (*PRA_mt0090*); however, none of these proteins were differentially expressed in response to promethazine (Appendix A). Together, this experiment demonstrates that promethazine induces a specific increase in extracellular laccase protein, and the changes in ABTS oxidation observed in this study are likely due to changes in laccase abundance. 

### 3.3. Promethazine Treatment Enhances the Ability of P. radiata to Degrade Phenolic Model Lignin Compounds

Laccases are versatile enzymes with diverse biological functions including the ability to catalyze the breakage of specific bonds found in lignin [23]. Because laccase activity increased when *P. radiata* was treated with promethazine, it was hypothesized that promethazine induction may enhance this organism’s ability to catalyze breakage in the types of bonds found in lignin. To test this hypothesis, *P. radiata* was treated with promethazine, and conditioned media samples (referred to as “secretomes”) were collected at different time points post treatment and tested for their abilities to degrade dimeric lignin model compounds (Figure 3A). Secretomes were collected at 2 h, 1 day, 2 days, and 7 days post promethazine treatment and were analyzed for their abilities to degrade established lignin model compounds 5,5′-dehydrodivanillate (DDVA), guaiacylglycerol-β-guaiacyl ether (GGE), syringylglycerol-β-guaiacyl ether (SGE), and veratrylglycerol-β-guaiacyl ether (VGE) using high-performance liquid chromatography (HPLC).

At 2 h post treatment, promethazine did not alter the ability of *P. radiata* to degrade any of the model lignin compounds tested. However, by 1 day post promethazine treatment, *P. radiata* secretomes exhibited an increased ability to degrade the phenolic dimers DDVA, GGE, and SGE likely due to increased laccase abundance. Promethazine-treated *P. radiata* secretomes collected at 2 days and 7 days post treatment also exhibited enhanced degradation of DDVA, GGE, and SGE. Interestingly, the secretomes collected 7 days post promethazine treatment had reduced ability to degrade DDVA compared to the corresponding 2-day time point. This reduction in DDVA transformation efficiency may be consistent with the reduced laccase activity observed at this 7-day time point in Figure 2. Finally, promethazine treatment did not alter the ability of *P. radiata* to degrade the non-phenolic VGE compound at any time points tested. Together, these results indicate that promethazine treatment enhances the ability of *P. radiata* to secrete enzymes that can transform phenolic, but not non-phenolic, substrates containing the β-*O*-4 linkage found in lignin molecules. 

### 3.4. Transcriptomic Analysis of the P. radiata Response to Promethazine

To better understand how *P. radiata* responded to promethazine at the molecular level and to help determine if the increase in laccase activity was a result of increased laccase expression, changes in gene expression were measured at three time points post promethazine treatment using RNA sequencing (RNA-seq). Changes in global transcription profiles over time in response to promethazine were first investigated using principal component analysis (PCA) (Figure 4A).

PCA revealed that at 2 h post promethazine addition, the treated and untreated samples clustered together, indicating at this time point the transcriptional profiles of both conditions were relatively similar. At this 2 h time point, the fewest number of differentially expressed genes were observed between promethazine-treated and untreated cells, with 20 upregulated and five downregulated in response to promethazine (Figure 4B). Among the most upregulated genes 2 h post promethazine treatment were laccase (*lac1*, gene model: plus.g7011.t1), a fungal hydrophobin (gene model: plus.g3107.t1), and an uncharacterized velvet factor-domain-containing protein (gene model: minus.g10711.t1). The most downregulated genes at the 2 h time point included a putative oxaloacetate acetylhydrolase (gene model: minus.g9799.t1), a lignin peroxidase precursor (gene model: minus.g3073.t1), and manganese peroxidase 3 (gene model: plus.g1419.t1). At 24 h, promethazine-treated and untreated samples exhibited a separation along the PC2 axis, suggesting that promethazine induced an altered transcriptional profile. These transcriptional results mirror the phenotypic data in Figure 3, demonstrating that by 24 h post promethazine treatment there is a change in transcriptional profiles and altered secretome properties. At 24 h post promethazine treatment, there was an increase in significantly differentially expressed genes with 46 genes upregulated and 34 genes downregulated in promethazine treated cells compared to non-treated controls. 

The phenotypes observed in this study associated with increased laccase activity and lignin-degrading properties suggested that promethazine may be inducing genes related to lignin-degradation. To test this hypothesis, genes that were induced in response to promethazine were queried for their molecular functions using GO term analysis (Figure 4C). Using this approach, 38 GO terms were assigned to genes within the set of differentially expressed genes. Interestingly, the most frequent GO term observed was “oxidoreductase activity” which appeared in 50% of the total hits. The second most observed GO term was “transferase activity” (~10%), followed by “hydrolase activity” (~5%). To better understand the nature of the genes being induced by promethazine, the most upregulated genes in response to the drug were investigated (Figure 4D). The most statistically significant upregulated gene in response to promethazine was a pleiotropic drug resistance ABC transporter (gene model: plus.g8349.t1). In addition, consistent with the previous experiments in this study, the second-most statistically significant upregulated gene in response to promethazine was laccase (gene model: plus.g7011.t1), which exhibited a 143× increase in expression compared to controls at 24 h post promethazine treatment. Other genes with functions associated with transformation of lignin or aromatic compounds were also upregulated in response to promethazine including a glutathione S-transferase (gene model: minus.g11347.t1) and the LigB extradiol aromatic ring-opening dioxygenase (gene model: plus.g3817.t1). Interestingly, at this 24 h time point, among the most significantly downregulated genes was a lignin peroxidase precursor (gene model: minus.g3073.t1), suggesting that the regulation of laccase and lignin peroxidase may be controlled by distinct factors. Similar to the 24 h time point, at 48 h post promethazine treatment the most upregulated genes were laccase, glutathione S-transferase, LigB aromatic ring opening dioxygenase, and a putative UDP-glycosyltransferase (gene model: plus.g696.t1). The most downregulated genes in promethazine-treated cells at 48 h post treatment were a hypothetical acetyltransferase (gene model: minus.g9894.t1), a hypothetical ARM repeat containing protein (gene model: plus.g4519.t2), and nuclease Le1 (gene model: minus.g10376.t1). Finally, laccase expression (gene model: plus.g7011.t1) was significantly upregulated in comparison to the controls at all time points examined post promethazine treatment: 13× by 2 h, 143× at 24 h, and 40× at the 48 h time point (Figure 4E). Additional laccase and laccase-like isoenzymes were not differentially expressed in response to promethazine. Together, these results indicate that laccase gene expression in *P. radiata* is rapidly induced in response to promethazine.

### 3.5. Promethazine Induction of Laccase Activity in Additional Wood-Degrading Fungi

The results for promethazine induction of laccase activity in *P. radiata* prompted the question of whether promethazine could induce laccase expression in other white-rot fungal species, or if this response was limited to *P. radiata*. To test this question, white-rot species *Bjerkandera adusta*, *Dichomitus squalens*, *Phanerochaete chrysosporium*, *Pleurotus ostreatus and Trametes versicolor* were treated with promethazine and measured for laccase activity in the media supernatants at different time points using ABTS oxidation assays (Figure 5 and Appendix A). Although *P. chrysosporium* does not encode laccase in its genome, it was included here as a negative control and as a reference due to its extensive examination as a model species. *Laetiporus gilbertsonii* was also included in these experiments as a wood-decay fungus that does not cause white rot. In these experiments, there was no significant change in laccase activity observed at any time point when *B. adusta*, *D. squalens*, *P. chrysosporium*, or *L. gilbertsonii* were treated with promethazine. Interestingly, 4 days post promethazine treatment, a significantly increased level of laccase activity was observed in *P. ostreatus* and two natural isolates of *T. versicolor*. At the 4-day time point, *P. ostreatus* exhibited a ~6× increase in laccase activity (*p* = 9 × 10^−6^, *t*-test), and the two *T. versicolor* isolates exhibited ~3× and ~5× increases in laccase activity when compared to untreated controls (*p* = 6 × 10^−4^ and 7 × 10^−4^, respectively; *t*-test). Together, these results indicate that promethazine treatment can induce laccase expression in multiple white-rot fungal species.

## 4. Discussion

In this study, phenothiazine compounds were found to rapidly induce laccase expression and lignin-degrading capabilities in the white rot fungus *P. radiata.* This report builds on previous studies that first identified culture conditions that promote laccase expression in *P. radiata,* including growing the organism on wood or supplementing growth media with copper, veratryl alcohol, or veratric acid [36,37,38,39,51]. Here, micromolar concentrations of the phenothiazine derivative promethazine caused upregulation of laccase gene expression within 2 h of treatment and increased extracellular laccase enzymatic activity by 24 h. Additionally, secretomes analyzed from promethazine-treated *P. radiata* exhibited increased abundance of laccase protein and an enhanced ability to degrade phenolic lignin-model compounds. Together, these studies reveal strategies to enhance lignin-degrading properties in *P. radiata* for biotechnological applications, and to decipher how the regulation of lignin-degrading enzyme expression may occur in nature.

Why might phenothiazines enhance lignin-degrading properties in *P. radiata?* Phenothiazines are sulfur-containing tricyclic organic molecules with numerous derivatives that are differentiated by diverse side chains extending from the middle ring moiety [52]. Although phenothiazines are most frequently used as psychotropic medications in humans, these compounds can also inhibit growth of fungal pathogens and have the potential to be repurposed as anti-fungal drugs [53,54,55,56]. The anti-fungal properties of phenothiazines primarily result from increases in the membrane permeability and disruption of the function of drug efflux pumps [55,56,57,58]. To our knowledge, this is the first study to investigate how phenothiazine compounds interact with and modulate the properties of wood-degrading fungi. Although the direct molecular mechanism regulating laccase induction in response to phenothiazines was not identified in this study, there are several potential mechanisms that could be investigated in future studies. One hypothesis is that the rapid induction of laccase after promethazine treatment is a stress response associated with plasma membrane disruption. As mentioned above, phenothiazines are well documented to alter the integrity of the plasma membrane in fungi. In fact, from the chemical screen performed in this study, several of the strongest inducers of laccase activity have known roles with altering membrane integrity or disrupting plasma membrane biosynthesis. For example, the strongest laccase inducer identified in this study was berberine chloride, and this compound has previously been demonstrated to have antifungal properties resulting from altering the integrity of the plasma and mitochondrial membranes [59,60]. Additionally, several azole compounds, including fluconazole, propiconazole, and miconazole, were found to induce laccase activity in *P. radiata.* These azoles are commonly used antifungal drugs and also affect the integrity of fungal membranes by inhibiting the biosynthesis of ergosterol, a key lipid responsible for maintaining membrane fluidity and permeability [61]. A second possibility is that upregulation of laccase is a response to promethazine-inhibiting drug efflux pumps at the cell surface. Previous studies have demonstrated that phenothiazines can strongly inhibit the function of ABC type efflux pumps in fungi [54,57]. Interestingly, in this study the most statistically significant upregulated gene in response to promethazine was a pleiotropic ABC-type drug resistance transporter, which may suggest that promethazine is interfering with the function of these efflux pumps. In either case, it is likely that upregulation of laccase in response to promethazine may be a defensive strategy to combat the toxic effects of this compound.

Promethazine-induced laccase activity was not limited to *P. radiata,* and similar responses were observed in the fungal species *P. ostreatus* and *T. versicolor.* Interestingly, a recent study demonstrated that emodin, a natural anthaquinone family compound with some structural similarity to phenothiazine, similarly increased laccase expression in *T. versicolor* [62]. The authors concluded that emodin may be inducing laccase expression in *T. versicolor* through activation of the xenobiotic response element (XRE) pathway as they observed upregulation of genes including laccase, cytochrome P450, and glutathione S-transferase (GST). *P. radiata* may similarly respond to promethazine through an XRE pathway as suggested by the upregulation of laccase, CYP450, and GST gene transcripts in response to promethazine observed in this study (see Figure 4). Although phenothiazines are synthetic compounds and are not known to exist in the environment, structurally similar phenothiazine-like compounds with antimicrobial properties are produced naturally by plants [58]. Therefore, it is possible that the rapid induction of laccase by *P. radiata* in response to phenothiazines may be a response that evolved to neutralize naturally occurring plant defense chemicals in the environment. However, it remains unclear by what molecular mechanism *P. radiata* uses to sense and respond to phenothiazines. 

Finally, the results presented here may be useful for the development of bioprocesses that involve utilization of laccase or the towards the valorization of lignin. Here, it was identified that promethazine can be utilized as a rapid and potent inducer of laccase expression and lignin-degrading properties in *P. radiata*. However, it remains unclear whether laccase by itself is capable of catalyzing bond breakage in high-molecular weight lignin polymers [63]. Although promethazine treatment enhanced the ability of *P. radiata* to degrade phenolic lignin model compounds, the effect that this treatment may have on *P. radiata’s* ability to modify polymeric lignin remains unknown. Additionally, the induction of laccase activity after promethazine treatment is transient and was observed to decline over time. This pattern of increasing laccase activity followed by a rapid reduction in activity is consistent with previous studies that investigated inducers of laccase in *P. radiata* [39]. For example, when *P. radiata* was treated with veratryl alcohol, laccase activity peaked ~3–5 days after treatment, and was followed by a rapid inactivation of the enzyme [38,39]. It remains unclear by what mechanism this inactivation of laccase activity occurs. Previous studies also observed that veratryl alcohol-treated *P. radiata* exhibited low levels of lignin peroxidase and manganese peroxidase activity at early time points post treatment and increased activity of these peroxidases after the laccase activity had declined [38,39]. Follow-up studies are needed to investigate the temporal dynamics of other lignin-degrading enzymes in response to promethazine and to further dissect the genetic regulation of these genes. Together, the experiments in this study identify phenothiazines as a class of chemical inducers of laccase expression in biotechnologically relevant fungal species.

## Figures and Tables

**Figure 1 jof-09-00371-f001:**
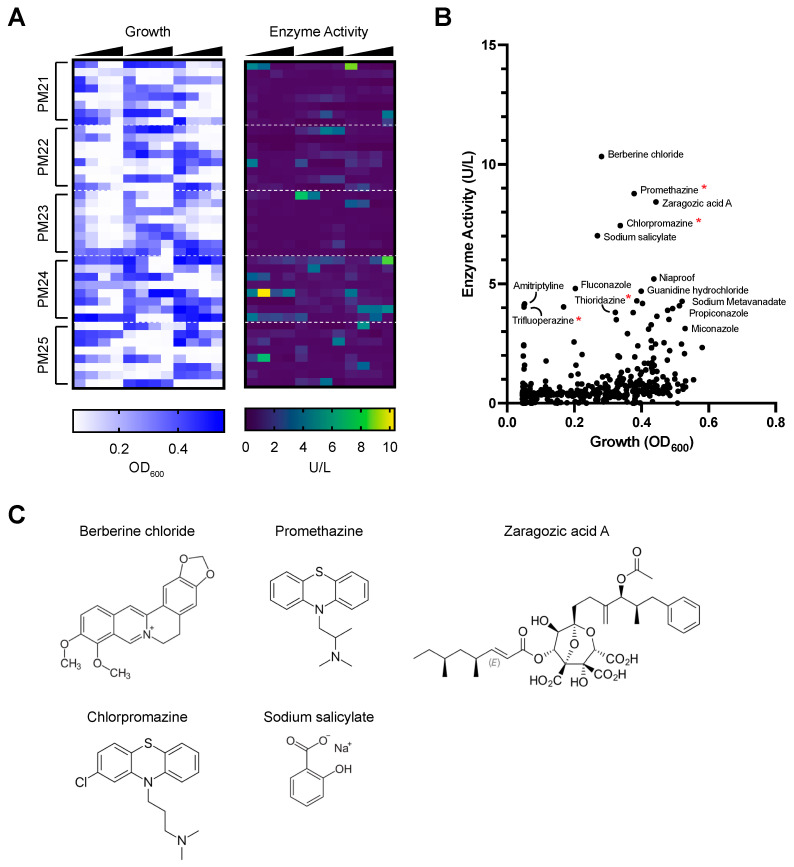
Chemical screen to identify inducers of laccase activity in *P. radiata*. (**A**) Heatmaps depicting growth of *P. radiata* using OD_600_ measurements (left) and extracellular enzyme activity using ABTS oxidation assays (right) in response to fungal chemical stressors in Biolog Phenotype Microarray (PM) plates. The heatmap orientation represents the 96-well plate layouts of the PM plates tested. Each chemical was tested across a range of four concentrations, grouped from low to high, and the concentration range is depicted by black triangles above the heatmap plots. Chemical names and their locations in the heatmaps can be found in Appendix A. Values depicted represent the average of *n* = 2 biological replicates. (**B**) Scatter plot depicting *P. radiata* growth vs. enzyme activity in the phenotype microarray chemical sensitivity screen. Red asterisks highlight phenothiazine compounds. (**C**) Chemical structures of the top 5 compounds that induced the highest levels of laccase activity in *P. radiata*.

**Figure 2 jof-09-00371-f002:**
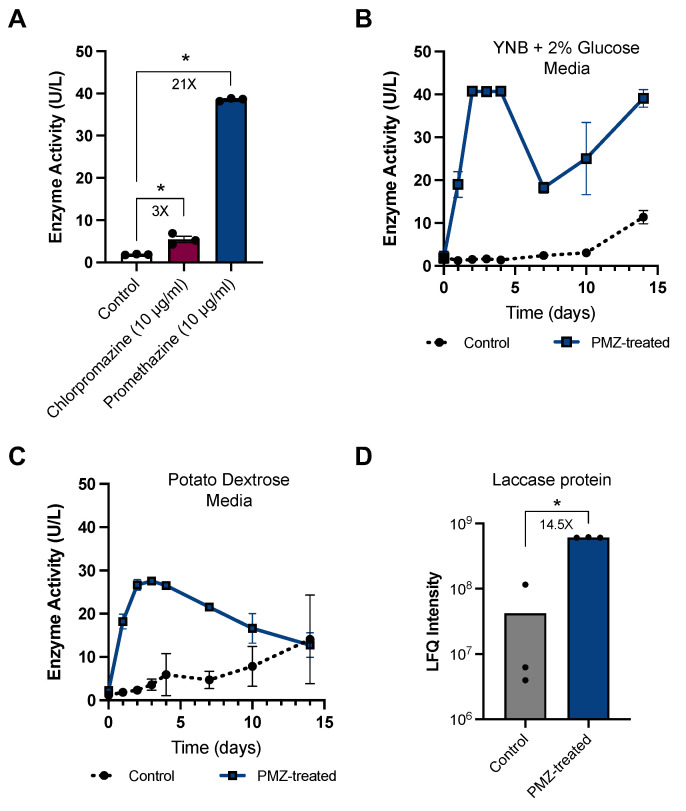
*P. radiata* laccase activity in response to promethazine treatment. (**A**) Scale-up laccase activity assays of *P. radiata* conditioned media after treatment with the top two phenothiazine compounds identified in the chemical screen: chlorpromazine and promethazine. *P. radiata* mycelia were grown in 20 mL YNB + 2% glucose and laccase activity was measured using ABTS oxidation assays at 2 days post treatment. Bars represent the mean of *n* = 3 biological replicates, and error bars represent the standard error of the mean. (**B**) Time course analysis of laccase activity in response to promethazine when *P. radiata* mycelia were cultured in YNB + 2% glucose or in (**C**) PD media. Each data point represents the mean of 2 biological replicates, and error bars depict standard deviation. (**D**) Quantification of laccase protein detected in *P. radiata* media supernatants 2 days post promethazine treatment using label-free quantitative proteomics analysis. Bars represent the mean of *n* = 3 biological replicates and individual data points are shown. Statistical significance is this figure is denoted by * for *p* < 0.01.

**Figure 3 jof-09-00371-f003:**
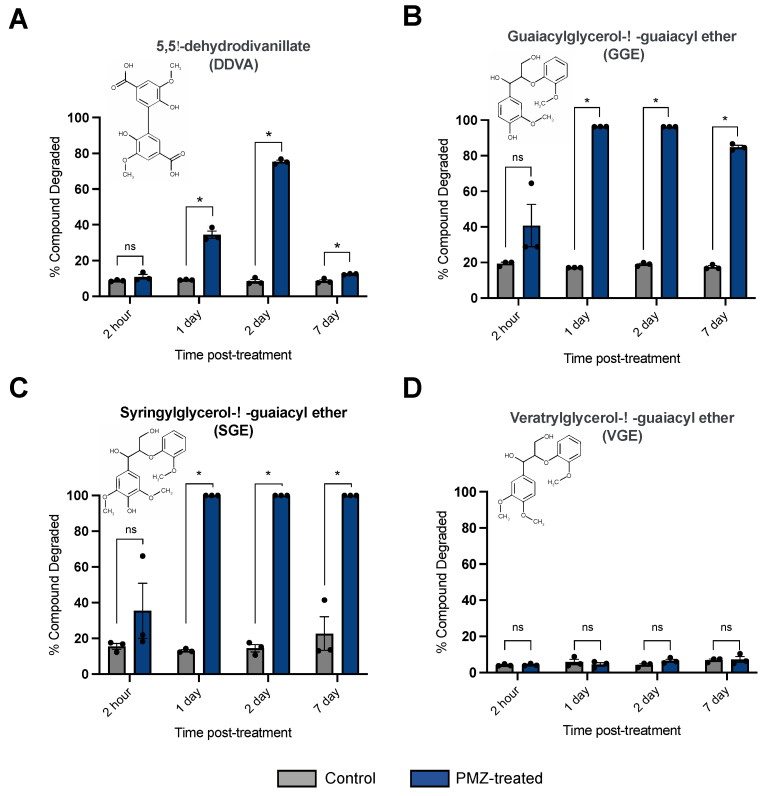
Functional assays of promethazine-induced *P. radiata* secretomes on dimeric lignin model compounds. Secretomes of *P. radiata* were collected at 2 h, 1 day, 2 days, and 7 days after promethazine treatment and then individually tested for the ability to degrade the dimeric lignin model compounds (**A**) 5,5′-dehydrodivanillate (DDVA), (**B**) guaiacylglycerol-β-guaiacyl ether (GGE), (**C**) syringylglycerol-β-guaiacyl ether (SGE), and (**D**) veratrylglycerol-β-guaiacyl ether (VGE). The lignin model compounds were incubated with *P. radiata* secretomes for 3 days at 25 °C, and then measured for degradation using HPLC peak-height calculations. Bars represent mean of 3 biological replicates, with individual data points shown, and error bars represent the standard error of the mean. Statistical significance is denoted by “ns” for not significant and * for *p* < 0.01.

**Figure 4 jof-09-00371-f004:**
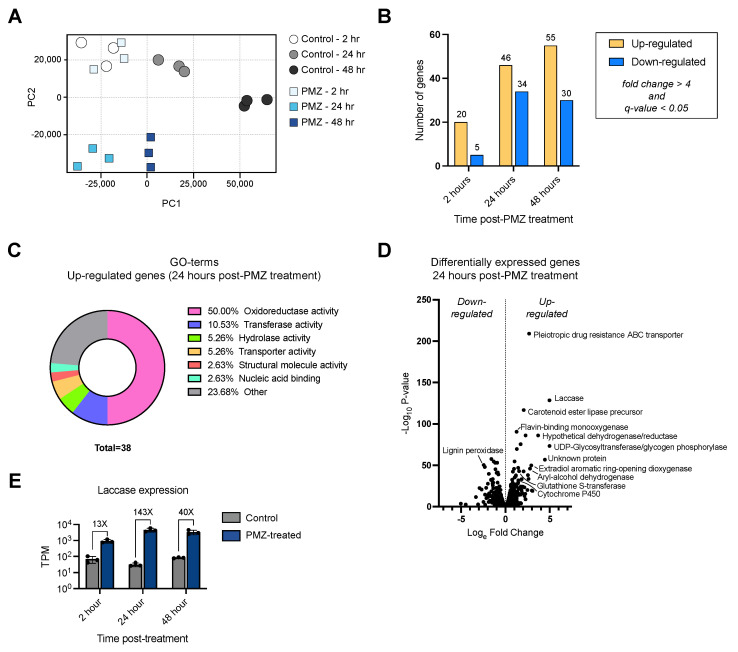
Transcriptomic analyses of *P. radiata* in response to promethazine treatment. (**A**) PCA plot depicting RNA-seq transcriptional profiles of untreated control and promethazine-treated *P. radiata* at different time points. (**B**) Bar chart of the number of significant differentially expressed genes at time points post treatment. Genes included here were filtered for a q-value < 0.05 and a fold change > 2. (**C**) Donut plot showing the breakdown of molecular function GO terms identified among the significantly upregulated genes 24 h after promethazine treatment. (**D**) Volcano plot highlighting a selection of the most upregulated genes in *P. radiata* 24 h after promethazine treatment. (**E**) Laccase gene expression levels measured at 2 h, 24 h, and 48 h post promethazine treatment depicted in transcripts per million (TPM). Bars represent the mean of three biological replicates with individual data points shown, and error bars depict the standard deviation.

**Figure 5 jof-09-00371-f005:**
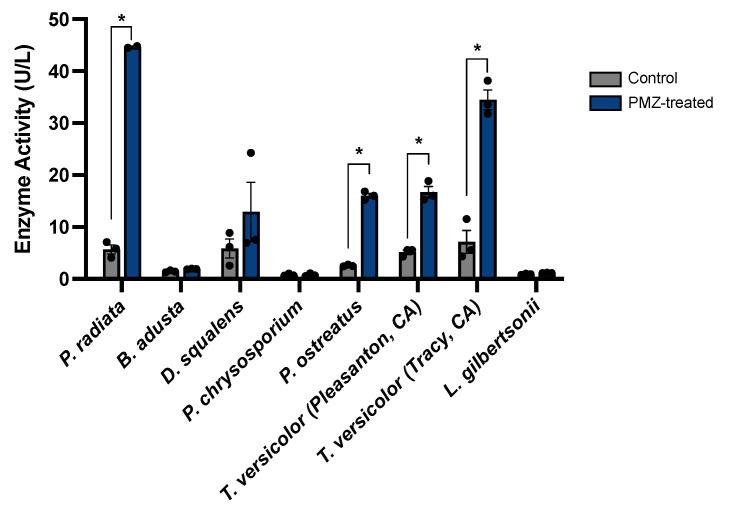
Promethazine induction of laccase activity in other wood-degrading fungal species. Laccase activity in media supernatants collected from diverse fungal species +/− promethazine treatment. Samples were analyzed for extracellular laccase activity at 4 days post treatment using ABTS assays (data from additional time points are displayed in Appendix A). Bars represent the mean of 3 biological replicates with individual data points shown, and error bars depict the standard error of the mean. Statistical significance is denoted as * for *p* < 0.01.

## Data Availability

RNA sequencing data were deposited into the National Center for Biotechnology Information Sequence Read Archive (SRA) and is available under the BioProject ID: PRJNA922187.

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
