# Peer review of "Phenothiazines Rapidly Induce Laccase Expression and Lignin-Degrading Properties in the White-Rot Fungus Phlebia radiata"

_jof, 2023, doi:10.3390/jof9030371_

Round 1
Reviewer 1 Report
The study is relevant to the field of the journal. The purpose of the study was stated well and achieved throughout the study. The methods used are adequate to perform work. The results obtained by the authors are original and important and are of interest to a wide range of journal readers. However, the paper shows several weak points listed below.
Materials and Methods section.
1. To measure laccase activity, the reaction mixture was incubated for 5 min and in some tests, E420 was up to 0.8. I am afraid that in these cases, during the 5-minute reaction, there was no direct relationship between the rate of oxidation of ABTS and time, and the authors gave underestimated results (in comparison with variants of the experiment with low enzymatic activity).
2. For comparison with literature data, the enzymatic activity must be expressed in international units.
3. Degradation reactions were incubated at 25°C for 3 days. Why three days? During incubation, at least partial inactivation of the enzyme occurred. Did the authors measure the residual activity of laccase in control and PMZ-treated mixtures?
Results
1. The main disadvantage of this work is that the authors did not evaluate the growth and biomass of the fungus grown under completely different conditions. Considering the compounds used and their concentrations, it is desirable to describe their possible effect (or lack thereof) on the growth and accumulation of fungal biomass.
2. P. 359. To “P. radiata secretomes exhibited an increased ability to degrade the phenolic dimers” must be added “due to higher laccase activity” since an increased ability to degrade the phenolic dimers correlates to the enzyme activity in samples.
3. It is not clear what the authors were guided by when choosing wood-rotting fungi, especially Phanerochaete chrysosporium which is not a laccase producer.
Reference 2 does not correctly cite the names of the authors.
Author Response
We thank this reviewer for taking the time to carefully review our manuscript and for their insightful comments. Their comments and suggestions along with our responses and edits to the manuscript have significantly improved the manuscript. Below are our point-by-point responses to the comments and suggestions made by the reviewer.
Comments and Suggestions for Authors
The study is relevant to the field of the journal. The purpose of the study was stated well and achieved throughout the study. The methods used are adequate to perform work. The results obtained by the authors are original and important and are of interest to a wide range of journal readers. However, the paper shows several weak points listed below.
Materials and Methods section.
- To measure laccase activity, the reaction mixture was incubated for 5 min and in some tests, E420was up to 0.8. I am afraid that in these cases, during the 5-minute reaction, there was no direct relationship between the rate of oxidation of ABTS and time, and the authors gave underestimated results (in comparison with variants of the experiment with low enzymatic activity).
We thank the reviewer for this observation and have added a line in the text mentioning that the E420 values may be underestimated (line 352-353).
- For comparison with literature data, the enzymatic activity must be expressed in international units.
We appreciate this suggestion to improve the manuscript. We have converted enzyme activity values from all enzyme activity assays into U/L and added a description of how units were calculated to the materials and methods section.
- Degradation reactions were incubated at 25°C for 3 days. Why three days? During incubation, at least partial inactivation of the enzyme occurred. Did the authors measure the residual activity of laccase in control and PMZ-treated mixtures?
We chose this incubation condition because it was intermediate to other degradation conditions reported in the literature (various temperatures, pHs, and incubation durations ranging from hours to days are reported). Because we used an unpurified, non-concentrated mixture of secreted enzymes, we wanted to ensure sufficient time for degradation to occur at a temperature that was physiologically relevant for the species. We did not test for residual activity of laccase activity at the end of the experiment, but agree it is possible that there may be some inactivation of the enzyme over time.
Results
- The main disadvantage of this work is that the authors did not evaluate the growth and biomass of the fungus grown under completely different conditions. Considering the compounds used and their concentrations, it is desirable to describe their possible effect (or lack thereof) on the growth and accumulation of fungal biomass.
We agree with the reviewer’s comment that it is important to document the effects of different compounds tested on fungal growth. We would like to point the reader to Figure 1A and 1B where growth measurements are shown for the chemical screen using OD600 measurements. We have also added a new supplementary data file that includes the raw growth and ABTS oxidation data for phenotype microarray experiments.
- P. 359. To “P. radiata secretomes exhibited an increased ability to degrade the phenolic dimers” must be added “due to higher laccase activity” since an increased ability to degrade the phenolic dimers correlates to the enzyme activity in samples.
We have added this statement to the text.
- It is not clear what the authors were guided by when choosing wood-rotting fungi, especially Phanerochaete chrysosporium which is not a laccase producer.
Phanerochaete chyrsosporium was included in Figure 5 as a reference because it is the most extensively studied white-rot species and served as a negative control for laccase activity in this experiment. We have added some text explaining why it was included despite not producing laccase (line 517-519).
References
Reference 2 does not correctly cite the names of the authors.
Reference 2 author names have been corrected.
Reviewer 2 Report
The manuscript under review presents a research on screening for laccase inductors among a wide range of compounds for white rot fungus Phlebia radiata. Selected compound – promethazine - was tested in a scaled-up experiment and transcriptional response of P. radiata during cultivation in presence of this compound was studied. Additionally, the degradation ability of the cultural broth was estimated with several dilignans. The manuscript describes novel findings in the field of laccase induction and transcription regulation in white-rot fungi and significance of the research is undeniable from both theoretical and applied point of view. However, some parts of the manuscript should be improved before publication.
Introduction: this part is well written and provides a sufficient background. But some of the cited publications are either out of date or irrelevant. I highly recommend to use up-to-date review articles wherever it is possible. Please, replace Ref 8-10 (L 43), Ref 25 (L72), Ref 27 (L69) with more relevant and up-to-date publications. Ref 21 does not support the information provided (L64).
Materials and Methods: in the paragraph “Chemical screen using Phenotype Microarrays” a nonconventional for basidiomycetes method of inoculum preparation is described. Please, provide corresponding references for this method. If this method was applied for the first time, it should be discussed. The same note is for PM application for basidiomycetes. For ascomycete fungi some researches modified the protocol (for example, comparing the readings at 590 nm and at 750 nm). Was there any optimization of the protocol in your case?
Results: To compare laccase induction by promethazine with other compounds 150 μM copper sulfate, 2 mM veratryl alcohol and 50 μg/ml lignin were used. What was the rationale for choosing these concentrations? Is it optimal for P. radiata? And why laccase activity was measured only at 2 day time point?
RNA-seq was performed at three time point. Data for 48 h time point almost not described. Please, provide the data (with GeneBank accessions and expression level for each gene) at least as supplementary materials. The GeneBank accessions should be added to the text when the individual genes are discussed. White-rot fungi possess several non-allelic gene copies for laccases and ligninolytic peroxidases. Please, add information about expression of other members of laccase and peroxidase gene families.
ABTS is not selective substrate for laccase activity. This should be kept in mind when you describe ABTS-activity in cultural broth. Please, revise L334-377. Also I should note that laccase ability for bond breaking in lignin is still under debates. You can find more information in [Munk, L., Sitarz, A. K., Kalyani, D. C., Mikkelsen, J. D., & Meyer, A. S. (2015). Can laccases catalyze bond cleavage in lignin?. Biotechnology advances, 33(1), 13-24]. Additionally, oxidative activity towards ABTS was found in P. chrysosporium cultural broth. It was called “laccase activity” but P. chrysosporium lacks laccase genes in its genome. Please, add some remarks in the manuscript about this fact. And I do not recommend using term “laccase activity” in case of P. chrysosporium. What was the rationale for choosing P. chrysosporium for laccase induction measurements?
As I understand from MM section for the degradation phenolic model lignin compound a filtered cultural broth containing living cells of P. radiata was used. I’m not sure that the term “secretome” can be used for it.
Promethazine induction of laccase activity in other wood-degrading fungal species was checked at 4 day. Why so? In different fungi laccase could appear at different time points. If there is an additional data on other time points, please, provide them.
Discussion: some information from second paragraph was repeated in the further text. It could be useful to add this information and corresponding references to the next paragraphs and here reduce it.
Minor remarks:
L45: please, use the term “izoenzymes” for same enzymes coded by different genes.
L121: “…phenotype micrarray experiments…” – misspelling
L323 and Supplementary Figure 3: please, add some additional explanation for curves in this figure. It is not absolutely clear, what was meant by “1 doze”, “2 doze” and “3 doze”. Was it three separate cultivations where PMZ was added as 1 doze, 2 and three sequential dozes?
L401: “…promethazine-induced…” - extra hyphen?
L423: “…and the LigB extradiol aromatic…” – misspelling
L466-467: could P. radiata encounter phenothiazine compound in nature? Please, add some additional remarks to these lines.
Author Response
We thank the reviewers for taking the time to carefully review our manuscript and for their insightful comments. Their comments and suggestions along with our responses and edits to the manuscript have significantly improved the manuscript. Below are our point-by-point responses to the comments and suggestions made by the reviewers.
Comments and Suggestions for Authors
The manuscript under review presents a research on screening for laccase inductors among a wide range of compounds for white rot fungus Phlebia radiata. Selected compound – promethazine - was tested in a scaled-up experiment and transcriptional response of P. radiata during cultivation in presence of this compound was studied. Additionally, the degradation ability of the cultural broth was estimated with several dilignans. The manuscript describes novel findings in the field of laccase induction and transcription regulation in white-rot fungi and significance of the research is undeniable from both theoretical and applied point of view. However, some parts of the manuscript should be improved before publication.
Introduction: this part is well written and provides a sufficient background. But some of the cited publications are either out of date or irrelevant. I highly recommend to use up-to-date review articles wherever it is possible. Please, replace Ref 8-10 (L 43), Ref 25 (L72), Ref 27 (L69) with more relevant and up-to-date publications. Ref 21 does not support the information provided (L64).
We would like to thank the reviewer for reading our manuscript and providing comments and suggestions to improve this manuscript. We have updated the references as requested and included more up-to-date references to the best of our knowledge.
Materials and Methods: in the paragraph “Chemical screen using Phenotype Microarrays” a nonconventional for basidiomycetes method of inoculum preparation is described. Please, provide corresponding references for this method. If this method was applied for the first time, it should be discussed. The same note is for PM application for basidiomycetes. For ascomycete fungi some researches modified the protocol (for example, comparing the readings at 590 nm and at 750 nm). Was there any optimization of the protocol in your case?
The protocol used in this study was adapted from protocols provided for Phenotype Microarrays by Biolog, and a previous study that conducted PM plate analysis on basidiomycetes. We did conduct some optimization to ensure that our inoculation method was standardized across all wells. We have added some discussion to the materials and methods section and included a reference where PM plates were used in Polypore basidiomycetes.
Results:
To compare laccase induction by promethazine with other compounds 150 μM copper sulfate, 2 mM veratryl alcohol and 50 μg/ml lignin were used. What was the rationale for choosing these concentrations? Is it optimal for P. radiata? And why laccase activity was measured only at 2 day time point?
We had originally chosen to show only the 2-day timepoint because it was the timepoint when we observed the highest levels of enzyme activity in response to promethazine. However, we agree with the reviewer that is informative to include additional timepoints to examine laccase induction in response to the other compounds. Therefore, we have included additional timepoints for this experiment at day 0, 2, 5 and 7 into a new supplementary figure (Supplementary Figure 4). We used concentrations that were within the range used in previous studies, but the concentrations, media, and duration may not be optimal for P. radiata. We have made a comment regarding this in the manuscript (line 376-379).
RNA-seq was performed at three time point. Data for 48 h time point almost not described. Please, provide the data (with GeneBank accessions and expression level for each gene) at least as supplementary materials. The GeneBank accessions should be added to the text when the individual genes are discussed. White-rot fungi possess several non-allelic gene copies for laccases and ligninolytic peroxidases. Please, add information about expression of other members of laccase and peroxidase gene families.
We have added some discussion of the 48h time point to the text and highlighted the most up- and down-regulated genes (line 496-502). We have also included the gene expression levels at different time points as a new supplementary data file. We have also added the gene annotation information for individual genes when discussed in the manuscript text. We also investigated expression levels of other laccase and laccase-like isoenzymes and found they were not differentially expressed in response to promethazine at any timepoint and added this to the text (line 505-506). For brevity, we have only highlighted the most significantly differentially expressed peroxidases.
ABTS is not selective substrate for laccase activity. This should be kept in mind when you describe ABTS-activity in cultural broth. Please, revise L334-377. Also I should note that laccase ability for bond breaking in lignin is still under debates. You can find more information in [Munk, L., Sitarz, A. K., Kalyani, D. C., Mikkelsen, J. D., & Meyer, A. S. (2015). Can laccases catalyze bond cleavage in lignin?. Biotechnology advances, 33(1), 13-24]. Additionally, oxidative activity towards ABTS was found in P. chrysosporium cultural broth. It was called “laccase activity” but P. chrysosporium lacks laccase genes in its genome. Please, add some remarks in the manuscript about this fact. And I do not recommend using term “laccase activity” in case of P. chrysosporium. What was the rationale for choosing P. chrysosporium for laccase induction measurements?
We thank the reviewer for highlighting that ABTS is not a selective substrate for laccase and providing an opportunity to expand on our study. We have therefore added new proteomics data from cultural broth that demonstrates the only significant differentially expressed protein in response to promethazine is laccase (Figure 2D and Supplementary Figure 5). We believe this data helps to support the findings that changes in ABTS oxidation observed in this study are due to increases in laccase expression.
We have also revised the recommended sentences to describe laccase as being capable of breaking the types of specific bonds found in lignin (rather than polymeric lignin as a whole) and provided a recent reference to support this statement (lines 398-399).
We have also added the reference by Munk et al. to the discussion and address the need to examine how promethazine-treated P. radiata may modify polymeric lignin in future work (lines 605-609).
We included P. chrysosporium as a reference species because it is the most extensively studied white-rot species. We have added a sentence rationalizing including it here despite the lack of laccase in the genome (lines 517-519). In the original version of this manuscript, enzyme activity was reported using raw absorbance measurements without background subtraction. As per the recommendation of another reviewer, all enzyme activity assays have been background subtracted and converted to Units / L. Using this approach, we do not see laccase activity from P. chrysosporium.
As I understand from MM section for the degradation phenolic model lignin compound a filtered cultural broth containing living cells of P. radiata was used. I’m not sure that the term “secretome” can be used for it.
We believe that the term secretome is appropriate for the context, but we have added a sentence defining what we mean by secretome to avoid any confusion (line 403).
Promethazine induction of laccase activity in other wood-degrading fungal species was checked at 4 day. Why so? In different fungi laccase could appear at different time points. If there is an additional data on other time points, please, provide them.
We appreciate this suggestion and agree that it is important to include data from additional timepoints because different fungal species may respond to promethazine at different rates. Therefore, we have included additional time-course data collected from multiple days post-promethazine treatment (day 0, 1, 2, 4 and 7) into a new supplementary figure (Supplementary Figure 6). We had originally shown only data from Day 4 because this was the timepoint when we observed differences in laccase activity across multiple species examined.
Discussion: some information from second paragraph was repeated in the further text. It could be useful to add this information and corresponding references to the next paragraphs and here reduce it.
We thank the reviewer for highlighting the redundancy in this section. We have consolidated and reduced the paragraphs that had overlapping information.
Minor remarks:
L45: please, use the term “izoenzymes” for same enzymes coded by different genes.
We have changed the word “isoforms” to “isoenzymes” in the text.
L121: “…phenotype micrarray experiments…” – misspelling
We have fixed this misspelling in the text.
L323 and Supplementary Figure 3: please, add some additional explanation for curves in this figure. It is not absolutely clear, what was meant by “1 doze”, “2 doze” and “3 doze”. Was it three separate cultivations where PMZ was added as 1 doze, 2 and three sequential dozes?
That is correct, these were separate cultivations where PMZ was added as 1, 2 or 3 doses. We have clarified the experimental setup in the Supplementary Figure 3 legend.
L401: “…promethazine-induced…” - extra hyphen?
We have removed the hyphen.
L423: “…and the LigB extradiol aromatic…” – misspelling
We believe this spelling is correct.
L466-467: could P. radiata encounter phenothiazine compound in nature? Please, add some additional remarks to these lines.
This is an interesting question, and we have added some additional remarks to address this question in the discussion section (line 594-599).
Round 2
Reviewer 1 Report
No comments.
Reviewer 2 Report
Thank you for detailed answers to my comments.